# Fair Domain Generalization with Arbitrary Sensitive Attributes

## Abstract

We consider the problem of fairness transfer in domain generalization. Traditional domain generalization methods are designed to generalize a model to unseen domains. Recent work has extended this capability to incorporate fairness as an additional requirement. However, it is only applicable to a single, unchanging sensitive attribute across all domains. As a naive approach to extend it to a multi-attribute context, we can train a model for each subset of the potential set of sensitive attributes. However, this results in $2^n$ models for $n$ attributes. We propose a novel approach that allows any combination of sensitive attributes in the target domain. We learn two representations, a domain invariant representation to generalize the model's performance, and a selective domain invariant representation to transfer the model's fairness to unseen domains. As each domain can have a different set of sensitive attributes, we transfer the fairness by learning a selective domain invariant representation which enforces similar representations among only those domains that have similar sensitive attributes. We demonstrate that our method decreases the current requirement of $2^n$ models to 1 to accomplish this task. Moreover, our method outperforms the state-of-the-art on unseen target domains across multiple experimental settings.

## 1 Introduction

The successful integration of AI into various sectors of society like healthcare, security systems, autonomous vehicles, and digital assistants, has led to profound impacts on the community. However, two significant concerns refrain us from trusting AI in critical applications. The *first* concern is distribution shift, where AI performs poorly when faced with data it hasn't seen before. Two popular frameworks that improve the generalization capability of a model to unseen data are domain adaptation (Long et al., 2015) and domain generalization (Blanchard et al., 2011). Domain adaptation techniques assume access to unlabelled samples of the target data while domain generalization has no exposure to target data. The *second* concern involves unwarranted biases in AI decisions. AI has exhibited biases in its decisions in the past, for example, predicting a higher risk of re-offense for African Americans compared to Caucasians (Tolan et al., 2019), biasing against Hispanics while assessing loan applications (Bartlett et al., 2022), and women while screening resumes (Cohen et al., 2019). Several methods (Chuang & Mroueh, 2021; Finocchiaro et al., 2021; Dwork et al., 2011; Kang et al., 2023) improve fairness by making models robust against various fairness metrics.

Models that cater to distribution shifts usually ignore fairness and vice versa. A few works (Rezaei et al., 2021; Stan & Rostami, 2023; Singh et al., 2021; Chen et al., 2022; Schumann et al., 2019; Pham et al., 2023) address distribution shift and fairness together. To the best of our knowledge, FATDM (Pham et al., 2023) is the only work that addresses fairness under domain generalization. However, FATDM assumes that only a single, fixed sensitive attribute exists in all domains including unseen domains (for example, the sensitive attribute is either race or gender, but not both). In reality, multiple sensitive attributes co-occur in the data. We may need to maintain fairness with respect to multiple sensitive attributes and these attributes need not be the same across all domains, especially in unseen target domains. The extension from single to multiple attributes in DG is not straightforward, as each domain may have a different set of sensitive attributes. When one attempts to overcome this by simply providing *extreme* level of fairness i.e. protecting against all sensitive attributes, model performance could unnecessarily be compromised given the inherent trade-off (Menon & Williamson, 2018) between fairness and accuracy. We illustrate this trade-off

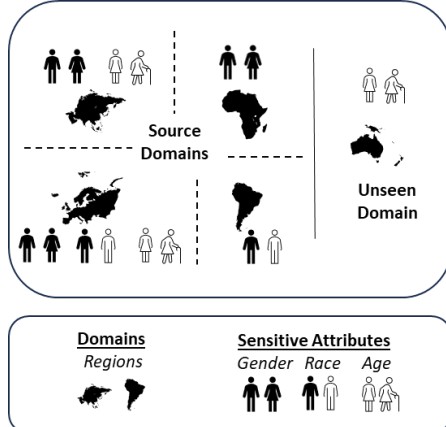

Figure 1: Fair DG with multiple sensitive attributes. A model trained with patient data from the source regions can be deployed in a new region with good performance and fairness with respect to all sensitive attributes for that region.

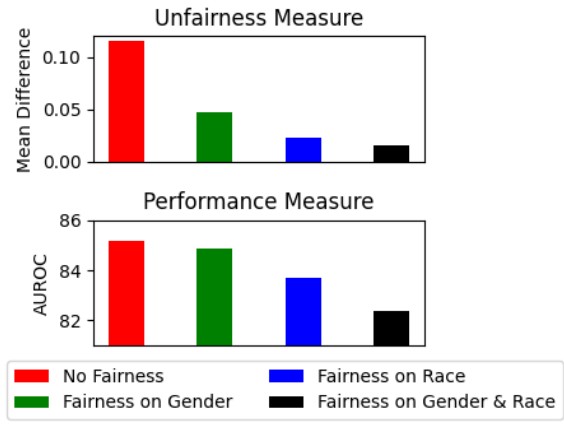

Figure 2: Performance-fairness trade-off in disease predictions with gender and race as sensitive attributes. We have performance (AUROC ↑) and unfairness metrics (Mean Difference ↓) from 4 models, each trained with a different level of fairness, tested on the same target domain. The drop in performance is high when fairness is enforced on multiple attributes.

in Fig 2. As another naive approach to extending to a multi-attribute setting, we can train different models for each combination of the sensitive attributes ($2^n$ models, for n sensitive attributes). However, the computational complexity of this approach scales exponentially with the number of sensitive attributes, making it an infeasible solution in most scenarios.

To overcome these limitations, we propose a framework that can adapt the model fairness based on the sensitive attributes of the unseen domain. Our proposed method extends the domain generalization methods to new dimensions where they not only generalize to tackle distribution shifts in the data but can also deal with any arbitrary subsets of sensitive attributes within a predefined set of sensitive attributes. Our novel problem of fair domain generalization with multiple sensitive attributes can be visualized in Figure 1. Our contributions are as follows:

- We introduce a new problem setting of fair domain generalization with multiple sensitive attributes. We are the first to devise a framework that can generalize across domains and maintain fairness transfer across multiple sensitive attributes to an unseen domain. Meanwhile, our model caters to the heterogeneous sensitivity of attributes across different domains and generalizes it to the target domain too.
- In order to tackle both problems, we learn two representations, one to encode generalization and the other, to encode fairness. Generalization representation is made domain invariant to generalize across domains with distribution shifts and fairness representation is made domain invariant only when their sensitive attributes match.
- We present a comprehensive training approach where we train the model with the concatenated representations of the source domain inputs and sensitivity information of the attributes. Compared to a naive approach which creates a model for each combination of the sensitive attributes, our method demonstrates a significant reduction in the number of models to be trained, reducing it from $2^n$ to just one, where $n$ is the total sensitive attributes.
- We conduct experiments on two real-world datasets and show that our method outperforms the state-of-the-art (Pham et al., 2023) in the presence of (2-4) sensitive attributes.

## 2 RELATED WORKS

**Domain Adaptation and Domain Generalization**   Domain adaptation (DA) Long et al. (2015); Ganin et al. (2015); Csurka (2017) techniques assume that unlabeled samples or in some cases few labeled samples of the target distribution are available for training. It is a framework to address

distribution shifts where we have some prior information on the target distribution. Under these assumptions, DA methods can yield some theoretical guarantees on their optimality on the source and the target distribution (Ben-David et al., 2010).

Domain generalization (DG) techniques (Blanchard et al., 2011) assume target domain data is unavailable during training. Instead, data is available from a set of distinct sources with some common characteristics. The aim of the model is to use multiple sources and generalize well to a target domain that is not present in the training set. DG techniques can be broadly categorized into domain invariant representation learning (Nguyen et al., 2021; Robey et al., 2021; Zhou et al., 2020; Arjovsky et al., 2019), data augmentation Volpi et al. (2018); Li et al. (2020); Gong et al. (2019) and various training strategies Mancini et al. (2018); Balaji et al. (2018). Studies for DA and DG focus only on generalization across domains and do not consider fairness.

**Fairness in Machine Learning**    Model fairness can be considered at an individual or a group level. Individual fairness requires similar individuals to be treated similarly (Mehrabi et al., 2022). Group fairness partitions the population into multiple groups and employs statistical measures across the groups to eliminate any bias towards certain groups. We address group fairness in this work. Most works consider the number of groups or sensitive attributes to be single. There are few works that specifically cater to multiple attributes (Kang et al., 2023; 2021). Group fairness methods can also vary based on the fairness measure used to equalize the groups. Commonly used group fairness measures are demographic parity (Dwork et al., 2011), equalized odds, and equal opportunity (Hardt et al., 2016). However, these methods assume that the training and testing distributions are identical.

**Fairness under Distribution Shift**    Recently, some works have proposed improving fairness in the context of distribution shifts. Schumann et al. (2019) attempt to transfer fairness from the sensitive attribute available during training (race) to an unseen sensitive attribute (gender). Singh et al. (2021) make assumptions on the data generative process based on causality and adds a fairness constraint to it to improve generalization and fairness on a target domain. Mandal et al. (2020) make their model robust against weighted perturbations of training data so that it is fair and robust with respect to a class of distributions. Chen et al. (2022) guarantee statistical group fairness when the distribution shift is within pre-specified bounds. An et al. (2022) maintain consistency across different sensitive attributes using self-training techniques. However, all these methods are under the domain adaptation setting where part of the target domain data is available during training.

Khoshnevisan & Chi (2021) develop a model that generalizes to an unseen sensitive attribute using only data from another attribute. However, their focus is on improving performance and they do not enforce any fairness measures. To the best of our knowledge, FATDM (Pham et al., 2023) is the first and only work that transfers accuracy and fairness to an unseen domain in a domain generalization setting. However, it works for a single sensitive attribute which is fixed across all the source domains. We can extend FATDM to a multi-attribute setting, either by strictly enforcing fairness on all sensitive attributes or by training different models for each combination of the sensitive attributes ($2^n$ models, for $n$ sensitive attributes). However, it either compromises model performance or is an infeasible solution when a large number of sensitive attributes are involved.

## 3    Proposed Framework

### 3.1    Problem Setting

We consider a novel setting of DG where data from multiple labeled source domains $\mathcal{T} = \{\mathcal{T}_1 \cup \mathcal{T}_2 \ldots \cup \mathcal{T}_m\}$ where $\mathcal{T}_d \sim \mathcal{P}_d$ is a collection of domain $d$ (where $d \in \mathcal{D}$) data of the form $(\mathbf{x}, y)$, where $\mathbf{x} \in \mathcal{X}$ is the input, $y \in \mathcal{Y}$ is the label of $\mathbf{x}$. The domains differ from each other due to covariate shift. The goal of the learner is to learn from $\mathcal{T}$ and obtain good predictive performance on a novel domain $\tilde{d}$ while maintaining adequate fairness across the sensitive attributes in $\tilde{d}$.

We use $\mathcal{S}$ to denote a predefined set of possible multiple sensitive attributes. We denote $n = |\mathcal{S}|$ as the number of sensitive attributes. However, each domain may have a subset of $\mathcal{S}$ as its set of sensitive attributes. The sensitivity configuration set is a power set of $\mathcal{S}$ and contains all possible configurations (subsets of $\mathcal{S}$) of the sensitive attributes a domain may take. For ease of representation, we map the configuration set to a corresponding set of binary vectors where 1 implies an

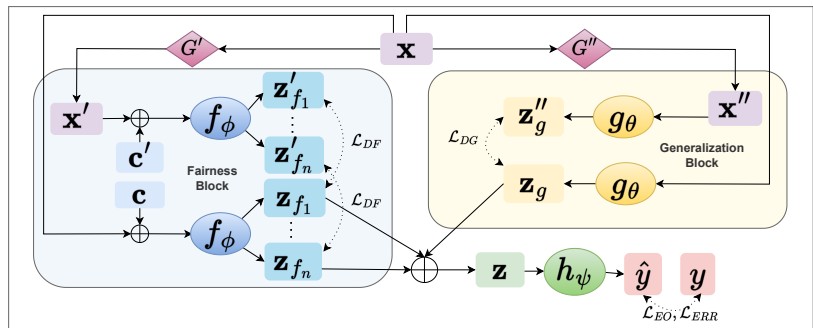

Figure 3: Overview of our approach. The block in blue represents the fairness components. The block in yellow represents the generalization components. The flow starts from $\mathbf{x}$ and ends at the model prediction $\hat{y}$. The inputs/outputs are in the round rectangles. The models that are part of the end-to-end training are in the ellipses.

attribute is sensitive and $0$ implies it is not. We denote the binary configuration set by $\mathcal{C}$. The cardinality of $\mathcal{C}$ is $2^n$. We use $\mathbf{c} \in \mathcal{C}, = (\mathbf{c}[0], \ldots, \mathbf{c}[n])$, where each $\mathbf{c}[i] \in \{0, 1\}$. Each sensitive attribute can take multiple instances. We use $\mathcal{I}_{\mathbf{c}}$ to denote the set of instances for a particular configuration $\mathbf{c}$ in $\mathcal{C}$. The following is an example when $n = 2$. $\mathcal{S} = \{\text{gender}, \text{race}\}$. The power set of $\mathcal{S} = \{[\text{none}], [\text{race}], [\text{gender}], [\text{gender}, \text{race}]\}$. The binary sensitivity configuration set, $\mathcal{C} = \{[0, 0], [0, 1], [1, 0], [1, 1]\}$. If $\mathbf{c} = [1, 1]$, the set of sensitive attribute instances for $\mathbf{c}$, $\mathcal{I}_{\mathbf{c}} = \{[\text{female}, \text{black}], [\text{female}, \text{white}], [\text{female}, \text{other}], [\text{male}, \text{black}], [\text{male}, \text{white}], [\text{male}, \text{other}]\}$. If $\mathbf{c} = [0, 1]$, then $\mathcal{I}_{\mathbf{c}} = \{[\text{black}], [\text{white}], [\text{other}]\}$.

## 3.2 METHOD

Our aim is to devise a framework that can transfer predictive performance and fairness from source domains to an (unseen) target domain. To this end, we propose to learn two domain invariant representations, one for transferring predictive performance and the other for transferring fairness of sensitive attributes. For transferring predictive performance across domains (generalization), we enforce the representations from different domains to be close when belong to the same class. For transferring fairness across domains, we enforce the representations from different domains to be close only if they have the same sensitive attributes. Otherwise, we increase the gap between the representations. Due to selectively applying invariance among the representations across domains, we refer to our method as *SISA: Selective Invariance for Sensitive Attributes*. Once both representations are learned, they are combined to get the final prediction. We describe the practical realization of our approach below. Figure 3 shows the overview of our approach.

### 3.2.1 MODULES

Our model has $4$ components.

**Domain Density Translators:** Represented by a mapping $G : \mathcal{X} \times \mathcal{D} \times \mathcal{D} \rightarrow \mathcal{X}$, $G$ translates a data point from one domain to another domain. That is, $G(\mathbf{x}, d, d')$ is intended to map a sample $\mathbf{x}$ of $d$ to a sample $\mathbf{x}' = G(\mathbf{x}, d, d')$ of domain $d'$. We use two such domain density translators, denoted by $G''$ and $G'$, to capture the domain-invariance representations for predictive performance and fairness, respectively. For domain generalization, we translate $\mathbf{x} \in d$ to $\mathbf{x}'' \in d'$ via a generative network $G''$. We enforce $G''$ to transform the data distribution within the class $y$ of $\mathbf{x}$.

$$\mathbf{x}'' = G''(\mathbf{x}, d, d') \tag{1}$$

Similarly, for fairness generalization, we translate $\mathbf{x}$ in domain $d$ to $\mathbf{x}'$ in domain $d'$ via a generative network $G'$. We enforce $G''$ to transform the data distribution within the class $y$ and the same sensitive attributes of $\mathbf{x}$.

$$\mathbf{x}' = G'(\mathbf{x}, d, d') \tag{2}$$

We train $G'$ and $G''$ as a precursor step through an optimization loss as discussed in Nguyen et al. (2021).

**Generalization Encoder:** $g_\theta : \mathcal{X} \to \mathcal{Z}_g$ encodes the data to an invariant representation that is generalizable across all domains. It is parameterized by $\theta$. $g_\theta$ receives $\mathbf{x}$ as its input and outputs the encoded representation $\mathbf{z}_g$.

$$\mathbf{z}_g = g_\theta(\mathbf{x}) \tag{3}$$

Given input $\mathbf{x}$ and its label $y$, we use Equation 1 to obtain $\mathbf{x}''$ from another domain $d' \sim \mathcal{D}$. Then we encode $\mathbf{x}''$ through $g_\theta$.

$$\mathbf{z}_g'' = g_\theta(\mathbf{x}'') \tag{4}$$

We learn $g_\theta$ via the following loss function.

$$\mathcal{L}_{DG} = \mathbb{E}_{d \sim \mathcal{D}} \mathbb{E}_{(\mathbf{x},y) \sim \mathcal{P}_d} \| \mathbf{z}_g - \mathbf{z}_g'' \|_2 \tag{5}$$

**Fairness Encoder:** $f_\phi : \mathcal{X} \times \mathcal{C} \to \{\mathcal{Z}_f\}_n$ encodes the data to a selectively invariant representation across the domains. It is parameterized by $\phi$. In our design, we consider $f_\phi$ to have $n$ outputs so that each sensitive attribute gets a corresponding fairness representation. However, we sometimes refer to the collection of $n$ representations as a single representation for better readability. In order to adapt the model to handle any sensitive attribute configuration $\mathbf{c} \sim \mathcal{C}$ during the test time, the input to $f_\phi$ should be the combinations of $\mathbf{x}$ with $\mathbf{c}$ ($\mathbf{x} \oplus \mathbf{c}$) $\forall \mathbf{c} \sim \mathcal{C}$. This approach might be cumbersome as it complicates the training of $f_\phi$ with $|\mathcal{C}|$ additional inputs per data point. In practice, we have found that randomly sampling a single $\mathbf{c} \sim \mathcal{C}$ and concatenating with $\mathbf{x}$ during each iteration was adequate for fairness transfer. So, the input to $f_\phi$ is a concatenation ($\oplus$) of $\mathbf{x}$ and a sensitive attribute configuration $\mathbf{c}$ randomly sampled from $\mathcal{C}$. $f_\phi$ outputs the representations $(\mathbf{z}_{f_1}, \ldots, \mathbf{z}_{f_n})$, where $\mathbf{z}_{f_i}$ corresponds to sensitive attribute $i$.

$$(\mathbf{z}_{f_1}, \ldots, \mathbf{z}_{f_n}) = f_\phi(\mathbf{x} \oplus \mathbf{c}) \tag{6}$$

To improve fairness to a target domain, we minimize the gap between the domain representations that have the same sensitive attribute configurations and maximize the gap for representations with different sensitive attributes. We sample another configuration $\mathbf{c}' \in \mathcal{C}$. In case of changes in the sensitive attributes, $\mathbf{z}_{f_i} = f(\mathbf{x} \oplus \mathbf{c})[i]$ and $\mathbf{z}'_{f_i} = f(\mathbf{x}' \oplus \mathbf{c}')[i]$ are made closer when $\mathbf{c}[i] = \mathbf{c}'[i]$ and far apart when $\mathbf{c}[i] \neq \mathbf{c}'[i]$. We realize this in practice using a contrastive loss and learn $f_\phi$ as below.

$$\mathcal{L}_{DF} = \begin{cases} \mathbb{E}_{d \sim \mathcal{D}} \mathbb{E}_{(\mathbf{x},y) \sim \mathcal{P}_d, d' \sim \mathcal{D}} \| \mathbf{z}_{f_i} - \mathbf{z}'_{f_i} \|_2 & \text{if } \mathbf{c}[i] = \mathbf{c}'[i] \ \ \forall\, i \in (1, n) \\ \mathbb{E}_{d \sim \mathcal{D}} \mathbb{E}_{(\mathbf{x},y) \sim \mathcal{P}_d, d' \sim \mathcal{D}} \max(0, \epsilon - \| \mathbf{z}_{f_i} - \mathbf{z}'_{f_i} \|_2) & \text{if } \mathbf{c}[i] \neq \mathbf{c}'[i] \ \ \forall\, i \in (1, n) \end{cases} \tag{7}$$

$\epsilon$ is a hyperparameter that controls the margins of the separation while maximizing the distance between the representations.

**Classifier:** $h_\psi : \mathcal{Z} \to \mathcal{Y}$ predicts the final output of the model. It is parameterized by $\psi$. The input to $h_\psi$ is the concatenation ($\oplus$) of the representations obtained from $g_\theta$ and $f_\phi$. It outputs the final prediction.

$$\hat{y} = h_\psi(\mathbf{z}_g \oplus \mathbf{z}_{f_1} \oplus \ldots \oplus \mathbf{z}_{f_n}) \tag{8}$$

Our classifier is trained with two objectives. The first one is to minimize the prediction error on the training domains (DG goal). The second one is to minimize the unfairness measure for the given set of sensitive attributes (Fairness goal).

We reduce the classification error between the true labels and the model predictions on the source domains. We concatenate the outputs of the encoders into the unified representation $\mathbf{z} = \mathbf{z}_g \oplus \mathbf{z}_{f_1} \ldots \oplus \mathbf{z}_{f_n}$. The classifier utilizes this representation as its input and subsequently produces the model prediction. We minimize $l : \mathcal{Y} \times \mathcal{Y} \to \mathbb{R}$, (e.g., cross entropy) between the model prediction and the true label.

$$\mathcal{L}_{ERR} = \mathbb{E}_{d \sim \mathcal{D}} \mathbb{E}_{(\mathbf{x},y) \sim \mathcal{P}_d} l(h_\psi(\mathbf{z}), y) \tag{9}$$

where $\mathbf{z}$ is computed using Equations (3), (6) and (8).

---

**Algorithm 1** *SISA:* Selective Invariance for Sensitive Attributes

---

**Require:** Training data: $\mathcal{T}$, sensitive attribute set $\mathcal{S}$, sensitive attribute configurations $\mathcal{C}$, density translators: $G''$, $G'$, batch size: $B$

1: Initialize $\theta$, $\phi$, and $\psi$, the parameters of $g_\theta$, $f_\phi$ and, $h_\psi$ respectively.
2: **for** epoch in MAX_EPOCHS **do**
3:    **for** each domain $d \in \mathcal{D}$ **do**
4:       Sample a batch $\{\mathbf{x}^k, y^k\}_{k=1}^{B} \sim \mathcal{P}_d$ from $\mathcal{T}$
5:       **for** each $k \in (1, B)$ **do**
6:          $\mathbf{z}_g^k \leftarrow g_\theta(\mathbf{x}^k)$                  *# Compute the generalization representation*
7:          $d' \sim \mathcal{D}$                        *# Sample another domain*
8:          $\mathbf{x}^{k''} \leftarrow G''(\mathbf{x}^k, d, d')$        *# Translate $\mathbf{x}$ into domain $d'$*
9:          $\mathbf{z}_g^{k''} \leftarrow g_\theta(\mathbf{x}^{k''})$          *# Compute the domain translated representation*
10:         $\mathbf{c} \sim \mathcal{C}$                       *# Sample a sensitive attribute configuration*
11:         $(\mathbf{z}_{f_1}^k, \ldots, \mathbf{z}_{f_n}^k) \leftarrow f_\phi(\mathbf{x}^k \oplus \mathbf{c})$    *# Compute the fairness representation*
12:         $\mathbf{x}^{k'} \leftarrow G'(\mathbf{x}^k, d, d')$
13:         $\mathbf{c}' \sim \mathcal{C}$                      *# Sample another sensitive attribute configuration*
14:         $\{\mathbf{z}_{f_1}^{k'}, \ldots, \mathbf{z}_{f_n}^{k'}\} \leftarrow f_\phi(\mathbf{x}^{k'} \oplus \mathbf{c}')$   *# Compute the domain translated representation*
15:         $\mathbf{z}^k \leftarrow \mathbf{z}_g^k \oplus \mathbf{z}_{f_1}^k \oplus \cdots \oplus \mathbf{z}_{f_n}^k$
16:       **end for**
17:       $\mathcal{L}_{DG} \leftarrow \sum_k [\| \mathbf{z}_g^k - \mathbf{z}_g^{k''}) \|_2]$          *# Invariance measure for performance*
18:       **for** each $i \in (1, n)$ **do**
19:          $\mathcal{L}_{DF} \leftarrow \mathcal{L}_{DF} + \sum_k [\mathbb{1}_{[\mathbf{c}[i] = \mathbf{c}'[i]]} \| \mathbf{z}_{f_i}^k - \mathbf{z}_{f_i}^{k'} \|_2 + \mathbb{1}_{[\mathbf{c}[i] \neq \mathbf{c}'[i]]} \max(0, \epsilon - \| \mathbf{z}_{f_i}^k - \mathbf{z}_{f_i}^{k'} \|_2)]$
20:       **end for**                       *# Selective invariance measure for fairness*
21:       $\mathcal{L}_{ERR} \leftarrow \sum_k [l(h_\psi(\mathbf{z}^k), y^k)]$       *# Classification loss over source domains*
22:       $\mathcal{L}_{EO} = \mathbb{E}_{d \sim D} \mathbb{E}_{(\mathbf{x}, y) \sim P_d} \mathbb{E}\_y \in Y \mathbb{E}\_i, j \in \mathcal{I}\_\mathbf{c}((h_\psi(\mathbf{z} \mid y, i) - (h_\psi(\mathbf{z} \mid y, j))^2$
23:       $\mathcal{L}_{final} \leftarrow \mathcal{L}_{ERR} + \omega \, \mathcal{L}_{EO} + \alpha \, \mathcal{L}_{DG} + \gamma \, \mathcal{L}_{DF}$
24:       $\theta \leftarrow \theta - \nabla_\theta \mathcal{L}_{final}$          *# Update network weights via gradient descent*
25:       $\phi \leftarrow \phi - \nabla_\phi \mathcal{L}_{final}$
26:       $\psi \leftarrow \psi - \nabla_\psi \mathcal{L}_{final}$
27:    **end for**
28: **end for**
29: **return** Trained $\theta, \phi, \psi$

---

We reduce the divergence in true positive rates and false positive rates across various instances of sensitive attributes. This measure is known as equalized odds (EO) and is popularly used in the fairness literature.

$$\mathcal{L}_{EO} = \mathbb{E}_{d \sim D} \mathbb{E}_{(\mathbf{x}, y) \sim P_d} \mathbb{E}\_y \in Y \mathbb{E}\_i, j \in \mathcal{I}\_\mathbf{c}((h_\psi(\mathbf{z} \mid y, i) - (h_\psi(\mathbf{z} \mid y, j))^2$$

where $\mathbf{z}$ is computed using Equations (3), (6) and (8).

Our final loss function is:

$$\mathcal{L}_{final} \leftarrow \mathcal{L}_{ERR} + \omega \, \mathcal{L}_{EO} + \alpha \, \mathcal{L}_{DG} + \gamma \, \mathcal{L}_{DF} \tag{10}$$

where $\alpha$, $\gamma$ and $\omega$ are hyperparameters. We train $g_\theta$, $f_\phi$, and $h_\psi$ together, end-to-end. We train density translators $G'$ and $G''$ prior to them. We summarize our proposed method in Algorithm 1.

## 4 EXPERIMENTS

We perform experiments to demonstrate the effectiveness of our proposed method. We first discuss the datasets and baselines used in our experiments, then we present the experimental results and model analysis.

### 4.1 DATASETS

**MIMIC-CXR** MIMIC-CXR (Johnson et al., 2019) contains chest-xrays of $227,827$ patients for various diseases. These images are linked with Johnson et al. (2021) dataset which includes patients'

Table 1: Details of target attributes, domains, and the sensitive attributes across the datasets.

| Dataset | Target Attribute (y) | | Domains (d) | | Sensitive Attribute ($\mathcal{S}$) | |
|---|---|---|---|---|---|---|
| | Name | Value | Name | Value | Name | Value |
| CelebA | Attractive | yes/no | hair color | brown/black/ blonde | big nose smiling male young | yes/no yes/no yes/no yes/no |
| MIMIC CXR | Cardiomegaly | yes/no | age | 0-40/40-60/ 60-80/80-100 | gender race | male/female white/black/other |
| | | | image rotations | 0°/15°/ 30°/ 45°/60° | gender race age | male/female white/black/other 0-40/40-60/ 60-80/80-100 |

information such as age, gender, race, etc. We chose the cardiomegaly disease prediction as our task. We created two different training sets $\mathcal{T}$ where for the first set, the domains were based on (i) the age of the patients, and for the second set, the domains were based on (ii) the degree of rotation of the images. We chose $\mathcal{S}$ as gender and race (set of sensitive attributes) for (i) and $\mathcal{S}$ as gender, race, and age for set (ii).

**CelebA** CelebA (Liu et al., 2015) contains $202,599$ face images from $10,177$ celebrities. Each of them is annotated with $40$ binary attributes based on facial features. This dataset is commonly used in fairness-based tasks. Our aim is to predict whether a given image is attractive or not. We create $3$ domains based on hair color. We chose the attributes big nose, smiling, young, and male as $\mathcal{S}$.

## 4.2 BASELINES

**DIRT:** We use a domain generalization algorithm DIRT (Nguyen et al., 2021) that uses domain density transformations between domains to create domain invariant representations. DIRT is FATDM without any fairness considerations.

**ERM:** We perform empirical risk minimization over the aggregated training data from all the domains. It is commonly used as a baseline in DG literature (Carlucci et al., 2019; Dou et al., 2019; Nguyen et al., 2021; Zhang et al., 2022).

**ERMF:** We perform empirical risk minimization over the training data but also employ the unfairness metric ($\mathcal{L}_{EO}$) to improve fairness.

**FATDM:** FATDM is the state-of-the-art (only) method that performs domain generalization and fairness to a target domain. However, FATDM considers only a single sensitive attribute that is fixed across domains. In order to compare with our method, we extend FATDM to a multi-attribute setting using a naive approach where we train a model for each configuration of the sensitive attribute. If there are $n$ sensitive attributes, we train $2^n$ models for FATDM.

## 4.3 IMPLEMENTATION DETAILS

Encoders $f$ and $g$ are Resnet-18 (He et al., 2016) for all the experiments. For a fair comparison, we used the same dimension for the representation $\mathbf{z}$ in our method and the baselines. The details are present in the appendix. Classifier $h$ is a single-layer dense network preceded by a ReLU layer activation for all the experiments. We modelled $G'$ and $G''$ using a StarGAN Choi et al. (2018). We chose hyperparameters $\alpha = 0.1$, $\omega = 1$ because FATDM also utilized these values for its hyperparameters, and they were found to be the most effective based on their reported results. We chose $\gamma = 1$ and $\epsilon = 1$ through hyperparameter tuning. We used Adam optimizer with a learning rate of $0.001$ for training.

Table 2: MIMIC-CXR dataset - Performance and fairness metrics averaged over the domains and the set of sensitive attribute configurations.

| Domains | Sensitive Attribute ($\mathcal{S}$) | Model | Performance Measures ($\uparrow$) | | | | Fairness Measures ($\downarrow$) | |
|---|---|---|---|---|---|---|---|---|
| | | | AUROC | AUPR | Acc | F1 | Mean | EMD |
| age | gender, race | ERM | 85.19 | 92.59 | 78.27 | 82.19 | 0.1153 | 0.2945 |
| | | DIRT | **85.20** | **92.62** | **78.69** | **82.65** | 0.1182 | 0.3078 |
| | | ERMF | 83.58 | 91.74 | 76.60 | 80.72 | 0.0298 | 0.1183 |
| | | FATDM (SOTA) | 83.66 | 91.85 | 76.05 | 81.08 | 0.0531 | 0.1248 |
| | | SISA (ours) | 84.71 | 92.35 | 76.99 | 81.89 | **0.0197** | **0.0873** |
| image rotations | gender, race, age | ERM | 81.61 | 91.46 | 74.57 | **81.09** | 0.2441 | 0.3907 |
| | | DIRT | 82.22 | 91.77 | 74.10 | 80.37 | 0.2785 | 0.4392 |
| | | ERMF | 81.75 | 91.70 | 73.86 | 80.15 | 0.0056 | 0.0547 |
| | | FATDM (SOTA) | 81.86 | 91.66 | 74.11 | 80.42 | 0.0045 | 0.0476 |
| | | SISA (ours) | **82.89** | **92.13** | **74.93** | **81.09** | **0.0022** | **0.0319** |

Table 3: CelebA dataset - Performance and fairness metrics averaged over the domains and the set of sensitive attribute configurations

| Domains | Sensitive Attribute ($\mathcal{S}$) | Model | Performance Measures ($\uparrow$) | | | | Fairness Measures ($\downarrow$) | |
|---|---|---|---|---|---|---|---|---|
| | | | AUROC | AUPR | Acc | F1 | Mean | EMD |
| hair color | big nose, smiling, male, young | ERM | **87.00** | **91.36** | **78.07** | **80.91** | 2.6214 | 1.2356 |
| | | DIRT | 86.92 | 91.30 | 77.94 | 80.63 | 3.0341 | 1.2039 |
| | | ERMF | 83.11 | 88.77 | 74.27 | 77.01 | 0.4248 | 0.3058 |
| | | FATDM (SOTA) | 83.02 | 88.75 | 74.12 | 76.77 | 0.2540 | 0.2533 |
| | | SISA (ours) | 84.82 | 90.02 | 75.86 | 78.67 | **0.0017** | **0.0195** |

## 4.4 EXPERIMENTAL RESULTS

**Experimental Setup:** We use the leave-one-domain-out evaluation scheme of domain generalization, where one of the domains is chosen as the target domain and kept aside for evaluation and the model is trained on the rest of the domains. We repeat this procedure over every domain in the dataset and average the results over them. We run the experiment 5 times with different seeds to report the mean results. As the sensitivity configuration of the target domain can be any $\mathbf{c} \in \mathcal{C}$, we calculate the performance and fairness obtained over each of them separately in Tables 10, 11, and 12 and combined in Tables 2 and 3, for MIMIC and CelebA datasets, respectively. The choice of domains, targets, and sensitive attributes for the two datasets are summarized in Table 1.

We consider several metrics to measure the performance and fairness of the model on the target domain. For performance, we consider AUROC (area under the receiver operating characteristic curve), AUPR (area under the precision-recall curve), Accuracy, and F1 score as the metrics that measure the performance of the model on the target domain. For fairness, we consider equalized odds as the unfairness metric and adopt Mean (distance between first moments of the distributions) and EMD (earth mover's distance) (Levina & Bickel, 2001) as the divergence measure Div$(. \parallel .)$.

**MIMIC-CXR:** For MIMIC-CXR, we have two sets of experiments considering, (i) the ages of the patients and (ii) the different rotations of the images as the domains for domain generalization. We consider gender and race to be sensitive attributes for (i) and gender, race, and age to be sensitive attributes for (ii). The prediction task is the existence of Cardiomegaly disease for (i) and (ii).

**CelebA:** For CelebA, we consider the various hair colors of celebrities as the domains of domain generalization. We consider big nose, smiling, male, and young attributes to be sensitive. The prediction task is whether a sample is attractive or not.

**Results:** We report the results of our experiments on MIMIC-CXR and CelebA in Tables 2 and 3 respectively. We average the predictive performance and report the best performances. Since ERM and DIRT do not optimize any fairness measures, they generalize fairness poorly. *Compared to the*

Table 4: Sensitivity analysis of parameter $\epsilon$ using CelebA dataset

| Model | $\epsilon$ | Performance Measures ($\uparrow$) | | | | Fairness Measures ($\downarrow$) | |
|---|---|---|---|---|---|---|---|
| | | **AUROC** | **AUPR** | **Acc** | **F1** | **Mean** | **EMD** |
| SISA | 0.01 | 83.73 | 88.82 | 74.65 | 77.73 | **0.0011** | **0.0148** |
| | 0.1 | 84.26 | 89.50 | 75.19 | 78.20 | 0.0020 | 0.0201 |
| | 1 | **84.82** | **90.02** | 75.86 | 78.67 | 0.0017 | 0.0195 |
| | 10 | 84.79 | 89.92 | **76.16** | **79.17** | 0.0045 | 0.0288 |

*methods that consider fairness, ERMF (baseline) and FATDM (state-of-the-art), our model SISA achieves the best values for performance and fairness on the target domain consistently on MIMIC-CXR and CelebA datasets.* Moreover, the predictive performance of SISA is not far off from ERM despite having to minimize a fairness constraint. We note an improvement of 1.05, 1.03, and 1.80 with a single model of SISA over 3, 8, and 15 models of FATDM for cases of 2, 3, and 4 sensitive attributes. The standard errors of our model and FATDM are reported in the appendix.

## 4.5 MODEL ANALYSIS

**Sensitivity analysis of $\epsilon$:**   $\epsilon$ is the hyperparameter that decides how apart $\mathbf{z}_{f_i}$ and $\mathbf{z}'_{f_i}$ should be if sensitive attribute $i$ is not equal. We train our model with $\epsilon = \{0.01, 0.1, 1, 10\}$ and report the results in Table 4. We do not observe a lot of difference in the performance and fairness as we changed $\epsilon$. In general, we notice slightly better results for performance when $\epsilon \geq 1$ and for fairness when $\epsilon = 0.01$. We chose $\epsilon = 1$ as it had the best predictive performance versus fairness trade-off.

**Sensitivity analysis of $\gamma$:**   $\gamma$ is the hyperparameter that decides the weight of $\mathcal{L}_{DF}$ loss in our model. We train our model with $\gamma = \{0.01, 0.1, 1, 10\}$ and report the results in Table 6. We do not observe a lot of difference in the performance and fairness as we changed $\gamma$. In general, we notice slightly better results for performance when $\gamma \uparrow$ and fairness when $\gamma \downarrow$. We chose $\gamma = 1$ as it had the best predictive performance versus fairness trade-off.

**Ablation study on the representations:**   To validate the efficacy of having two separate encoders to model the domain shift (generalization) and fairness, we conduct a study where we only use a single encoder to model both. We report the results in Table 7 in the appendix. We find that having multiple (two) encoders to model the representations improved the predictive performance while a single encoder improved the fairness.

## 5 CONCLUSION

We introduced a new problem setting of fair domain generalization with multiple sensitive attributes. We developed a framework that can generalize across domains and maintain fairness transfer across multiple sensitive attributes to an unseen domain. We learned two representations, one to encode generalization and the other, to encode fairness, and used their combined strength to get a model that is fair to any specified sensitive attribute and has a good predictive performance on the target domain. Our model outperformed the state-of-the-art FATDM in fair domain generalization over both performance and fairness. Our work is the first to tackle the challenging setting of fair domain generalization with multiple sensitive attributes. Currently, our model caters to arbitrary subsets within a prespecified sensitive attribute set. In the future, it would be interesting to extend it to completely unseen sensitive attributes in a zero-shot learning manner.

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

Table 5: Dimension of $\mathbf{z}$

| Model | MIMIC ($n=2$) $\mathbf{z}$ | | MIMIC ($n=3$) $\mathbf{z}$ | | CelebA $\mathbf{z}$ | |
|---|---|---|---|---|---|---|
| | $\mathbf{z}_g$ | $\mathbf{z}_{f_i}$ | $\mathbf{z}_g$ | $\mathbf{z}_{f_i}$ | $\mathbf{z}_g$ | $\mathbf{z}_{f_i}$ |
| ERM | | 1024 | | 1280 | | 1024 |
| DIRT | | 1024 | | 1280 | | 1024 |
| FATDM | | 1024 | | 1280 | | 1024 |
| SISA | 512 | 256 | 512 | 256 | 768 | 64 |

## A  APPENDIX

**Fairness Bound**   From FATDM paper, we have:

$$\epsilon_{D^T}^{EO}(\hat{f}) \leq \frac{1}{N} \sum_{i=1}^{N} \epsilon_{D^{S_i}}^{EO}(\hat{f}) + \sqrt{2} \min_{i \in [N]} \sum_y \sum_a d_{JS}(P_{D^T}^{X|Z=y,A=a}, P_{D^{S_i}}^{X|Y=y,A=a}) \tag{11}$$

$$+ \sqrt{2} \max_{i,j \in [N]} \sum_y \sum_a d_{JS}(P_{D^{S_i}}^{Z|Z=y,A=a}, P_{D^{S_j}}^{Z|Y=y,A=a}) \tag{12}$$

Unfairness measure within a domain $D_i$:

$$\epsilon_{D^{S_i}}^{EO}(\hat{f}) = \mathbb{E}_{D_{S_i}}[\hat{f}(X)_1 \mid Y=1, A=\mathbf{a}] - \mathbb{E}_{D_{S_i}}[\hat{f}(X)_1 \mid Y=1, A=\mathbf{a}'] + \tag{13}$$

$$\mathbb{E}_{D_{S_i}}[\hat{f}(X)_1 \mid Y=0, A=\mathbf{a}] - \mathbb{E}_{D_{S_i}}[\hat{f}(X)_1 \mid Y=0, A=\mathbf{a}'] \tag{14}$$

$$R_{D_{S_i}}^{y,\mathbf{a}} = \mathbb{E}_{D_{S_i}}[\hat{f}(X)_1 \mid Y=y, A=\mathbf{a}] \tag{15}$$

$$R_{D_{S_i}}^{y,\mathbf{a}} = \mathbb{E}_{D_{S_i}}[\hat{f}(X)_1 \mid Y=y, A=\mathbf{a}] \tag{16}$$

$R_{D_{S_j}}^{y,\mathbf{a}}$ can be bounded by $R_{D_{S_i}}^{y,\mathbf{a}}$ as follows provided, they have the same $A$?

**Dimension of $\mathbf{z}$**   We report the dimensions used for the representation $\mathbf{z}$ in Table 5. We consider the same dimensions for $\mathbf{z}$ across the baselines and our model for a valid comparison. We use the same dimension value for each $\mathbf{z}_{f_i}$'s.

**Results with standard errors**   We report the standard errors of FATDM and our model in Tables 8, 9. In most cases, the standard errors for performance metrics are less than 0.001, for both FATDM and our model. Standard errors for fairness metrics are less than 0.001 for Mean and 0.05 for EMD. In general, we observed that our model exhibits a lower standard error than FATDM.

**Results with all configurations of sensitive attributes in target**   We report the results for all sensitive attribute configurations in Tables 10, 11, 12. Additionally, for SISA we also report the results for None attribute where we do not consider any fairness attributes. However, we did not include this result while averaging to get a fair comparison with the FATDM baseline as FATDM does not have this configuration.

**Ablation study on the representations**   We report the results of the study on the number of representations used in Table 7.

Table 6: Sensitivity analysis of parameter $\gamma$ using CelebA dataset

| Model | $\gamma$ | Performance Measures ($\uparrow$) | | | | Fairness Measures ($\downarrow$) | |
|---|---|---|---|---|---|---|---|
| | | AUROC | AUPR | Acc | F1 | Mean | EMD |
| SISA | 0.01 | 84.44 | 89.88 | 75.42 | 78.25 | **0.0009** | **0.0165** |
| | 0.1 | 84.36 | 89.72 | 75.17 | 77.89 | 0.0016 | 0.0194 |
| | 1 | **84.82** | **90.02** | **75.86** | 78.67 | 0.0017 | 0.0195 |
| | 10 | 84.76 | 89.98 | 75.85 | **78.78** | 0.0014 | 0.0216 |

Table 7: Ablation on the number of encoders used

| Model | Number of Encoders Used | Performance Measures ($\uparrow$) | | | | Fairness Measures ($\downarrow$) | |
|---|---|---|---|---|---|---|---|
| | | AUROC | AUPR | Acc | F1 | Mean | EMD |
| SISA | one | 82.36 | 87.82 | 73.47 | 76.10 | 0.0003 | 0.0109 |
| | two | 84.82 | 90.02 | 75.86 | 78.67 | 0.0017 | 0.0195 |

Table 8: MIMIC-CXR dataset - Performance and fairness metrics averaged over the domains and the set of sensitive attribute configurations with standard errors.

| Domains | Sensitive Attribute ($\mathcal{S}$) | Model | Performance Measures ($\uparrow$) | | | | Fairness Measures ($\downarrow$) | |
|---|---|---|---|---|---|---|---|---|
| | | | AUROC | AUPR | Acc | F1 | Mean | EMD |
| age | gender | FATDM (SOTA) | 83.66 | 91.85 | 76.05 | 81.08 | 0.0531 | 0.1248 |
| | | **Standard error** | 0.0017 | 0.0009 | 0.0039 | 0.0044 | 0.0070 | 0.0134 |
| | race | SISA (ours) | 84.71 | 92.35 | 76.99 | 81.89 | **0.0197** | **0.0873** |
| | | **Standard error** | 0.0019 | 0.0017 | 0.0038 | 0.0039 | 0.0030 | 0.0110 |
| image rotations | gender | FATDM (SOTA) | 81.86 | 91.66 | 74.11 | 80.42 | 0.0045 | 0.0476 |
| | | **Standard error** | 0.0280 | 0.0290 | 0.0256 | 0.0270 | 0.0035 | 0.0108 |
| | race, age | SISA (ours) | **82.89** | **92.13** | **74.93** | **81.09** | **0.0022** | **0.0319** |
| | | **Standard error** | 0.0036 | 0.0021 | 0.0045 | 0.0044 | 0.0007 | 0.0050 |

Table 9: CelebA dataset - Performance and fairness metrics averaged over the domains and the set of sensitive attribute configurations with standard errors.

| Domains | Sensitive Attribute ($\mathcal{S}$) | Model | Performance Measures ($\uparrow$) | | | | Fairness Measures ($\downarrow$) | |
|---|---|---|---|---|---|---|---|---|
| | | | AUROC | AUPR | Acc | F1 | Mean | EMD |
| hair color | big nose, smiling, | FATDM (SOTA) | 83.02 | 88.75 | 74.12 | 76.77 | 0.2540 | 0.2533 |
| | | **Standard error** | 0.0033 | 0.0026 | 0.0055 | 0.0075 | 0.0560 | 0.0316 |
| | male, young | SISA (ours) | 84.82 | 90.02 | 75.86 | 78.67 | **0.0017** | **0.0195** |
| | | **Standard error** | 0.0053 | 0.0030 | 0.0039 | 0.0046 | 0.0005 | 0.0031 |

Table 10: MIMIC (i) Performance and Fairness with all configurations in Target domain

| Model | Target Sensitive Attributes | Performance Measures $\uparrow$ | | | | Fairness Measures $\downarrow$ | |
|---|---|---|---|---|---|---|---|
| | | AUROC | AUPR | Acc | F1 | Mean | EMD |
| FATDM gender | gender | 84.86 | 92.57 | 76.84 | 81.78 | 0.0947 | 0.2385 |
| FATDM race | race | 83.71 | 91.98 | 75.79 | 80.94 | 0.0231 | 0.0893 |
| FATDM gender race | gender, race | 82.40 | 91.01 | 75.91 | 80.26 | 0.0064 | 0.0522 |
| SISA | gender | 84.78 | 92.49 | 76.77 | 81.73 | 0.0554 | 0.2117 |
| | race | 84.78 | 92.47 | 77.17 | 82.03 | 0.0035 | 0.0362 |
| | gender, race | 84.58 | 92.09 | 77.03 | 81.92 | 0.0003 | 0.0142 |
| | None | 84.70 | 92.47 | 76.80 | 81.76 | - | - |

Table 11: MIMIC (i) Performance and Fairness with all configuration in Target domain

| Model | Target Sensitive Attributes | Performance Measures ↑ | | | | Fairness Measures ↓ | |
|---|---|---|---|---|---|---|---|
| | | AUROC | AUPR | Acc | F1 | Mean | EMD |
| FATDM gender | gender | 83.73 | 92.60 | 76.03 | 82.15 | 0.0143 | 0.1340 |
| FATDM race | race | 83.02 | 92.42 | 75.14 | 81.37 | 0.0054 | 0.0730 |
| FATDM age | age | 82.23 | 92.12 | 73.87 | 80.17 | 0.0108 | 0.0693 |
| FATDM gender race | gender, race | 81.28 | 91.36 | 73.85 | 80.22 | 0.0006 | 0.0281 |
| FATDM gender age | gender, age | 81.50 | 91.75 | 73.03 | 79.34 | 0.0005 | 0.0145 |
| FATDM race age | race, age | 80.98 | 90.92 | 73.86 | 80.27 | 0.0001 | 0.0106 |
| FATDM gender race age | gender, race, age | 80.26 | 90.42 | 72.97 | 79.43 | 0.0000 | 0.0033 |
| | gender | 83.60 | 92.64 | 75.60 | 81.74 | 0.0118 | 0.1129 |
| | race | 83.15 | 92.34 | 75.66 | 81.81 | 0.0019 | 0.0429 |
| | age | 82.49 | 92.20 | 74.50 | 80.69 | 0.0016 | 0.0419 |
| SISA | gender, race | 82.92 | 92.00 | 75.35 | 81.51 | 0.0001 | 0.0100 |
| | gender, age | 82.78 | 92.13 | 74.76 | 80.95 | 0.0001 | 0.0072 |
| | race, age | 82.71 | 91.92 | 74.98 | 81.16 | 0.0000 | 0.0057 |
| | gender, race, age | 82.59 | 91.68 | 73.66 | 79.77 | 0.0000 | 0.0025 |
| | None | 83.64 | 92.65 | 75.66 | 81.81 | - | - |

Table 12: MIMIC (i) Performance and Fairness with all configuration in Target domain

| Model | Target Sensitive Attributes | Performance Measures ↑ | | | | Fairness Measures ↓ | |
|---|---|---|---|---|---|---|---|
| | | AUROC | AUPR | Acc | F1 | Mean | EMD |
| FATDM big nose | big nose | 86.41 | 91.02 | 77.02 | 79.62 | 0.5399 | 0.6242 |
| FATDM smiling | smiling | 86.60 | 91.09 | 77.48 | 80.02 | 0.1768 | 0.3710 |
| FATDM male | male | 85.49 | 90.42 | 75.96 | 78.81 | 1.6107 | 1.1951 |
| FATDM young | young | 84.91 | 90.20 | 76.10 | 78.61 | 0.8258 | 0.8864 |
| FATDM big nose smiling | big nose, smiling | 83.43 | 89.09 | 75.18 | 78.16 | 0.0779 | 0.1008 |
| FATDM big nose male | big nose, male | 82.83 | 88.45 | 73.67 | 76.69 | 0.0542 | 0.0790 |
| FATDM big nose young | big nose, young | 81.66 | 87.76 | 73.54 | 76.12 | 0.1003 | 0.1078 |
| FATDM smiling male | smiling, male | 83.56 | 89.23 | 72.63 | 75.20 | 0.1761 | 0.1441 |
| FATDM smiling young | smiling, young | 81.95 | 88.21 | 73.59 | 75.93 | 0.0603 | 0.0928 |
| FATDM male young | male, young | 81.89 | 88.31 | 72.58 | 74.85 | 0.1632 | 0.1150 |
| FATDM big nose smiling male | big nose, smiling, male | 82.05 | 87.93 | 72.52 | 75.01 | 0.0017 | 0.0179 |
| FATDM big nose smiling young | big nose, smiling, young | 81.98 | 88.09 | 74.08 | 76.92 | 0.0024 | 0.0181 |
| FATDM big nose male young | big nose, male, young | 80.26 | 86.53 | 72.26 | 75.20 | 0.0165 | 0.0239 |
| FATDM smiling male young | smiling, male, young | 80.80 | 87.51 | 71.90 | 74.36 | 0.0033 | 0.0180 |
| FATDM big nose smiling male young | big nose, smiling, male, young | 81.46 | 87.39 | 73.20 | 76.04 | 0.0003 | 0.0058 |
| | big nose | 85.14 | 90.41 | 76.34 | 79.32 | 0.0035 | 0.0439 |
| | smiling | 86.27 | 90.87 | 77.08 | 80.31 | 0.0006 | 0.0313 |
| | male | 83.88 | 89.44 | 74.97 | 77.98 | 0.0160 | 0.1046 |
| | young | 84.59 | 90.18 | 75.61 | 78.46 | 0.0031 | 0.0578 |
| | big nose, smiling | 85.47 | 90.31 | 76.64 | 79.62 | 0.0002 | 0.0057 |
| | big nose, male | 84.63 | 89.89 | 75.88 | 78.80 | 0.0007 | 0.0071 |
| | big nose, young | 84.95 | 90.28 | 76.18 | 79.05 | 0.0002 | 0.0064 |
| SISA | smiling, male | 84.53 | 89.80 | 75.57 | 78.45 | 0.0005 | 0.0080 |
| | smiling, young | 85.41 | 90.46 | 76.42 | 79.34 | 0.0002 | 0.0071 |
| | male, young | 84.09 | 89.80 | 75.36 | 78.08 | 0.0006 | 0.0077 |
| | big nose, smiling, male | 84.81 | 89.79 | 75.72 | 78.44 | 0.0001 | 0.0025 |
| | big nose, smiling, young | 84.96 | 89.94 | 76.07 | 78.72 | 0.0000 | 0.0025 |
| | big nose, male, young | 84.51 | 89.84 | 75.67 | 78.24 | 0.0001 | 0.0031 |
| | smiling, male, young | 84.43 | 89.66 | 74.92 | 77.39 | 0.0000 | 0.0025 |
| | big nose, smiling, male, young | 84.57 | 89.59 | 75.43 | 77.83 | 0.0000 | 0.0018 |
| | None | 86.10 | 90.79 | 76.91 | 80.22 | - | - |

Table 13: Pearson Correlation between Model Outputs and Sensitive Attributes.

| Dataset | Domain | Pearson Correlation | | | |
|---|---|---|---|---|---|
| MIMIC-CXR | Image Rotation | Gender | | Race | Age |
| | ERM | 0.027 | | 0.007 | 0.177 |
| | SISA | 0.016 | | 0.000 | 0.103 |
| CelebA | | Big Nose | Male | Smiling | Young |
| | ERM | -0.186 | 0.188 | -0.383 | 0.332 |
| | SISA | -0.136 | 0.239 | -0.329 | 0.257 |

