# OpenReview forum: "Fair Domain Generalization with Arbitrary Sensitive Attributes"
_ICLR.cc/2024/Conference — Submitted to ICLR 2024_

### Official Review · Reviewer_cyJi · 2023-10-24

**Soundness:** 2 fair
**Presentation:** 3 good
**Contribution:** 2 fair
**Rating:** 3
**Confidence:** 4

**Summary:**

This paper studied a novel fair domain generalization problem where multiple sensitive attributes existed in different domains. The key challenge of this fair domain generalization problem is to deal with multiple potential sensitive attributes and any combinations of sensitive attributes can appear in the unseen testing domains. Then it presented a feasible solution by learning the domain-invariant representation and sensitive attribute invariant representation from training domains. The objective function included four components: domain-invariance loss, fairness-aware invariance loss, classification loss, and equalized odds (for fairness) loss. Experiments on two real-world data sets showed that the proposed outperformed DG baselines in terms of generalization and fairness.

**Strengths:**

**Originality:** This paper focused on a novel fair domain generalization problem with multiple sensitive attributes. It was a much more challenging problem setting than previous work due to the complicated interconnections of different sensitive attributes. The major technical novelty of this paper was to selectively learn the invariant representation based on the sensitive attributes, e.g., generate representations with respect to sensitive attributes. Experimental results demonstrated the effectiveness of the proposed SISA method over several baselines in terms of both generalization and fairness metrics.

**Quality:** The fair domain generalization problem was well-defined. The motivating example in Figure 1 also illustrated that fair domain generalization with multiple sensitive attributes was a challenging yet practical problem. In the derived objective function, both generalization performance and fairness were encouraged in different loss terms.

**Clarity:** Overall, the presentation of this paper was clear and the derived problem was well-motivated. Experiments also showed the training procedures and evaluation metrics for performance comparison between the proposed method and baselines. Ablation studies supported that the hyper-parameters were relatively robust to the model performance.

**Significance:** This paper extended previous fair domain generalization to a more general setting where multiple sensitive attributes could appear in different domains.

**Weaknesses:**

**W1:** The technical novelties of this paper are unclear. The proposed SISA approach involves several techniques from previous works, e.g., invariant representation learning with domain density translators, equalized odds loss, contrastive loss, etc. The major technical contributions could be emphasized in the context of the derived fair domain generalization problem.

**W2:** The explanation of the fairness encoder in subsection 3.2.1 is not convincing. (1) It randomly chooses a single $c$ to learn the representations of sensitive attributes. This can lead to biased and unstable solutions. More empirical evaluations on this sampling strategy can be provided. (2) It uses the concatenation between input $x$ and attribute $c$. However, if $x$ (e.g., high-dimensional images) and $c$ differ in dimensionality, would the concatenated vector be dominated by one of them?

**W3:** The equalized odds loss $\mathcal{L}_{EO}$ is confusing. What are $h\_{\psi}(z | y, i)$ and $h\_{\psi}(z | y, j)$ over the sample $(x, y) \sim \mathcal{P}\_d$? What is $\mathcal{P}$ within $Div(\cdot)$? Why does it involve both $\mathbb{E}\_{(x,y)\sim \mathcal{P}_d}$ and $\mathbb{E}\_{y \sim \mathcal{Y}}$?

**W4:** The hyper-parameter setting is not explained. (1) It shows that $\alpha=0.1, \omega= 1$ are used in previous work for the best reported results, and thus those parameters are also used in this paper.  However, since different models are involved in this paper and previous work, $\alpha=0.1, \omega= 1$ might lead to sub-optimal solutions. (2) The hyper-parameter sensitivity on $\epsilon$ and $\gamma$ are analyzed. However, it is unclear whether the best hyper-parameters are selected based on the testing domains. Is any validation method adopted for hyper-parameter selection during training?

**Questions:**

Q1: Figure 2 shows that the drop in performance is high when fairness is enforced on multiple attributes. This might indicate that it becomes more challenging to find the trade-off between generalization performance and fairness when increasing the number of sensitive attributes. Therefore, it would be better to provide some insights into understanding how to balance generalization performance and fairness when a large number of sensitive attributes exist.

Q2: Table 7 shows that the number of encoders can also affect the trade-off between performance and fairness. Why does a single encoder improve the fairness and multiple encoders help generalization performance in the proposed approach?

Q3: This paper considers the covariate shift among domains. Can the proposed SISA method be adapted to deal with other types of distribution shifts, e.g., label shifts, concept shifts, etc?

########################################

After reviewing the rebuttals, I would like to keep my rating unchanged, since most of my concerns have not been addressed. In most cases, the responses are not very convincing. More theoretical or empirical results could be added to support the explanations, e.g., the selection of $c$, hyper-parameter selection, the trade-off between performance and fairness, etc.

---

> ### Author Response · Authors · 2023-11-22
> **Rebuttal for Reviewer 4: Part 1**
>
> **We thank Reviewer 4 for their constructive and appreciative comments about our paper. We address their queries below.**
>
> > 1) The technical novelties of this paper are unclear. The proposed SISA approach involves several techniques from previous works. The major technical contributions could be emphasized in the context of the derived fair domain generalization problem.
>
> We thank Reviewer 4 for the suggestion and have revised our paper elucidating our contribution better.
>
> ***
> >2) a) Fairness encoder randomly chooses a single $\mathbf{c}$ to learn the representations of sensitive attributes. This can lead to biased and unstable solutions. More empirical evaluations on this sampling strategy can be provided.
>
> + Initially, we had trained the model on all $\mathbf{c}$ $\in \mathcal{C}$ during each iteration. We found that it increased the training complexity. Then, we sampled a $\frac{1}{3}$ number of  $\mathbf{c}$'s from $\mathcal{C}$ during each training iteration and found that it did not compromise the stability of the training. Even though in the algorithm we wrote it as sampling a single $\mathbf{c}$ in each iteration, in practice we sampled multiple $\mathbf{c}$'s in each iteration eventually reaching $\frac{1}{3}$ of the total elements in $\mathcal{C}$. We provide our code for a better understanding of our algorithm.
>
> > b)  Fairness encoder uses the concatenation between input $\mathbf{x}$  and attribute $\mathbf{c}$. However, if $\mathbf{x}$ and $\mathbf{c}$ differ in dimensionality, would the concatenated vector be dominated by one of them?
>
> + $\mathbf{c}$ is a binary vector of size $n$x$1$. We reshape it to $256$x$256$x$1$ by repeating its values over dimensions 1 and 2 and adding a 3rd dimension. Then it is concatenated to input $\mathbf{x}$ as an additional channel. So the total dimension of the input becomes ($\mathbf{x} + \mathbf{c} = $ $256$x$256$x$2$ for MIMIC CXR and $256$x$256$x$4$ for CelebA). $\mathbf{c}$ will not dominate $\mathbf{x}$ as variation between different $\mathbf{c}$'s is much lower than variations between different $\mathbf{x}$'s.
>
> ***
>
> > 3) The equalized odds loss $L_{EO}$ is confusing. What are $h_\psi(\mathbf{z} \mid y,i)$ and $h_\psi(\mathbf{z} \mid y,j)$ over the sample $(\mathbf{x},y) \sim P_d$? What is P within Div()? Why does it involve both $\mathbb{E}_{(\mathbf{x},y)} \sim P_d$ and $\mathbb{E}_y \sim Y$?
>
> $\mathbf{z}$ denotes the latent representation of $\mathbf{x}$ obtained through the fairness encoder $f_\phi$ and generalization encoder $g_\theta$ (Eq 3 and 6 from the paper). Hence even though the operation is on $\mathbf{z}$ inside $h_\psi$, it is a function of the training data $\mathbf{x}$ which is sampled from $(\mathbf{x},y) \sim P_d$. We have simplified $L_{EO}$ for better readability below.
>
> $L_{EO} = \mathbb{E}\_{d \sim D} \mathbb{E}\_{(\mathbf{x}, y) \sim P_d}\mathbb{E}\_{y \in Y} \mathbb{E}\_{i,j \in \mathcal{I}\_\mathbf{c}} ((h_\psi(\mathbf{z} \mid y, i) - (h_\psi(\mathbf{z} \mid y, j))^2$
>
> ***
>
> > 4) a) The hyper-parameter setting shows that $\alpha=0.1, \omega=1$ are used in previous work for the best-reported results, and thus those parameters are also used in this paper. However, since different models are involved in this paper and previous work, $\alpha=0.1, \omega=1$ might lead to sub-optimal solutions.
>
> We chose to use the best-reported hyperparameter values $\alpha=0.1, \omega=1$ for both the baseline and our model in order to have a fair comparison between the models. We understand the comments of the reviewer about tuning $\alpha=0.1, \omega=1$ for the baselines and our models, but we are unable to do so at the moment. However, we would like to highlight that despite tuning for the best values of $\alpha=0.1, \omega=1$, we were still able to achieve good performance from our model for the given values of hyperparameters.
>
> > b) The hyper-parameter sensitivity on $\epsilon$ and $\gamma$ are analyzed. However, it is unclear whether the best hyper-parameters are selected based on the testing domains. Is any validation method adopted for hyper-parameter selection during training?
>
> The hyperparameter values are chosen based on the highest accuracy on the validation set. However, for the ablation studies, we have reported their respective accuracies obtained on the test set.

---

> ### Author Response · Authors · 2023-11-22
> **Rebuttal for Reviewer 4: Part 2**
>
> > 5) Figure 2 shows that the drop in performance is high when fairness is enforced on multiple attributes. This might indicate that it becomes more challenging to find the trade-off between generalization performance and fairness when increasing the number of sensitive attributes. Therefore, it would be better to provide some insights into understanding how to balance generalization performance and fairness when a large number of sensitive attributes exist.
>
> ###### Our ablation studies in Tables 4, 5, and 6 of the paper show that our model's hyperparameters $\epsilon$ and $\gamma$ can be varied to manage the trade-off between fairness and accuracy quantitatively. For CelebA dataset, with the highest no of sensitive attributes ($4$), we noticed slightly better results for performance when $\gamma$ and $\epsilon$ values are high and fairness when $\gamma$ and $\epsilon$ are low.
>
> ***
>
> > 6) Table 7 shows that the number of encoders can also affect the trade-off between performance and fairness. Why does a single encoder improve the fairness and multiple encoders help generalization performance in the proposed approach?
>
> ###### In the case of a single-encoder model, a single representation $\mathbf{z}$ denotes the fairness and the generalization information. Hence, it is implicitly equally divided among the loss for the $n$ sensitive attribute ($L_{DF}$) and the generalization loss ($L_{DG}$). As there are $n$ sensitive attributes, it overshadows the generalization information due to being the same representation.
>
> ###### In the case of two-encoders model, where one encoder stands for fairness and the other for generalization performance, $\mathbf{z}$ is explicitly split between $\mathbf{z}_g$ and $\mathbf{z}_f$, giving $\mathbf{z}_g$ a good enough representation in $\mathbf{z}$ and not get overshadowed by $\mathbf{z}_f$. Hence, the generalization performance (accuracy) is better with two encoders.
>
> ***
>
> > 7) This paper considers the covariate shift among domains. Can the proposed SISA method be adapted to deal with other types of distribution shifts, e.g., label shifts, concept shifts, etc?
>
> ###### We have only considered covariate shift as the distribution shift in the scope of this paper. We plan to extend our work to other types of shifts in the future.

---

### Official Review · Reviewer_XuYi · 2023-10-29

**Soundness:** 2 fair
**Presentation:** 2 fair
**Contribution:** 2 fair
**Rating:** 3
**Confidence:** 4

**Summary:**

This paper introduces an approach aimed at achieving intersectional fairness within the context of domain generalization. Specifically, the proposed method focuses on acquiring two distinct invariant representations across domains, emphasizing both accuracy and fairness. Subsequently, a classifier is employed to make predictions based on these representations. To transfer fairness and accuracy into new domains, the authors train the model to minimize error and fairness loss across source domains.

**Strengths:**

- This paper targets fairness which is an important research topic in machine learning.

**Weaknesses:**

- The clarity of the paper is lacking, with several important details omitted. For instance, the paper lacks comprehensive information about the training of the domain density translator $G'$ for fairness generalization. Training $G'$ is not straightforward due to the varying sensitive attributes across different datasets.

- The design choice of using a shared translator $G'$ for all sensitive attributes appears questionable. Notably, given an input $X$ from domain $d$, $G'$ only generates $X' = G'(X, d, d')$ in domain $d'$, without considering which sensitive attributes are relevant to the translation. This implies that the model assumes $P_d(X|y,s) = P_{d'}(X'|y,s)$ and $P_d(X|y,s') = P_{d'}(X'|y,s')$ for every $s, s' \in S$, which is a strong assumption and may not hold in practical scenarios.

- The rationale behind learning distinct representations for each sensitive attribute is not well elucidated. Why is it necessary for the model to minimize the gap between domain representations with the same sensitive attribute configurations while maximizing the gap for those with different sensitive attributes? How does concatenating all representations contribute to accurate and fair predictions in target domains?

- The results in Table 2 and 3 seem to be presented for a fixed value of $\gamma$. It would be more comprehensive if the authors varied $\gamma$ to explore the accuracy-fairness trade-off for the methods used in the experiments.

- The final objective, as defined in Equation (10), encompasses a blend of multiple loss components. I recommend that the authors carry out an ablation study, varying hyperparameters, to assess the impact of each loss on the model's performance.

- The technical novelty of the paper appears somewhat limited. The primary contribution appears to be the utilization of distinct representations for each sensitive attribute.

- In the introduction section (Fig. 1), the authors assert that the proposed method can accommodate the heterogeneity of sensitive attributes across domains. However, in the experimental section, the models seem to have access to all sensitive attributes in all domains, which may contradict the initial claim.

- The paper lacks the provision of code and supplementary documentation, which could significantly enhance clarity and reproducibility. Providing these resources would be beneficial for the reader to understand and replicate the methodology.

**Questions:**

Please see the Weaknesses.

---

> ### Author Response · Authors · 2023-11-22
> **Rebuttal for Reviewer 3 : Part 1**
>
> **We thank Reviewer 3 for their constructive comments to improve our paper. We post responses to their queries below.**
>
> > 1) The clarity of the paper is lacking, with several important details omitted. For instance, the paper lacks comprehensive information about the training of the domain density translator $G'$ for fairness generalization. Training $G'$ is not straightforward due to the varying sensitive attributes across different datasets.
>
> We have added more details about the training of $G'$ in the supplementary section of the revised version of the paper.
>
> ***
> > 2) The design choice of using a shared translator $G'$ for all sensitive attributes appears questionable. Notably, given an input $X$ from domain $d$, $G'$ only generates $X'=G'(X,d, d')$ in domain $d'$, without considering which sensitive attributes are relevant to the translation. This implies that the model assumes $P_d(X\mid d,s) = P_{d'}(X'\mid d',s)$ and $P_d(X \mid d,s') = P_{d'}(X'\mid d,s')$ for every $s,s' \in S$, which is a strong assumption and may not hold in practical scenarios.
>
> We follow the design choice of $G'$ from FATDM who have assumed the same assumption you have stated and have proved with theoretical bounds that it does not compromise the fairness metric measures.
>
> ***
>
> > 3) a)The rationale behind learning distinct representations for each sensitive attribute is not well elucidated.
>
> We learn distinct representations for each sensitive attribute due to two reasons:
> 1. If there exists a representation for a sensitive attribute S, and if two domains A and B have S as a sensitive attribute, then one can seamlessly minimize the distance between the representations of two domains A and B that only pertain to S and not affect the representations of other sensitive attributes in A and B.
> 2. If there exists a representation for a sensitive attribute S, and if two domains A and B do not share its sensitivity, then one can seamlessly maximize the distance between representations of A and B that only pertain to S.
>
> > b) Why is it necessary for the model to minimize the gap between domain representations with the same sensitive attribute configurations while maximizing the gap for those with different sensitive attributes?
>
> When the sensitive attribute configurations are the same, that is, a domain A has its sensitive attribute set as gender and race and a domain B has its sensitive attribute set as gender and race, we want the predictions of both A and B to be equalized across all values taken by gender and race. Hence, we minimize the distance between the representations that pertain to fairness ($\mathbf{z}_f$) of domain A and domain B. \
> On the other hand, if the sensitive attribute configurations are different, that is domain A has its sensitive attribute set as gender and domain B has its sensitive attribute set as race, then the sensitive attribute set is different, we do not want them to have a similar prediction and maximize the distance between $\mathbf{z}_f$'s of domains A and B
>
> > c) How does concatenating all representations contribute to accurate and fair predictions in target domains?
>
> The representation $\mathbf{z}\_f$ is optimized for better fairness through loss $L_{DF}$ and the representation $\mathbf{z}\_g$ is optimized for better accuracy through loss $L_{DG}$. We use concatenation as it is an easy operation that can combine the two representations to get a final representation which is then sent to the classifier for a prediction that is fair and accurate.
>
> ***
>
> > 4) The results in Tables 2 and 3 seem to be presented for a fixed value of $\gamma$. It would be more comprehensive if the authors varied $\gamma$ to explore the accuracy-fairness trade-off for the methods used in the experiments.
>
> We have chosen $\gamma$ from hyperparameter tuning and have reported the results for the best $\gamma$ in Tables 2 and 3 in the paper. We have provided an ablation study for $\gamma$ in Table 5 and show that varying $\gamma$ has an effect on the accuracy-fairness trade-off.
>
> ***
>
> > 5) The final objective, as defined in Equation (10), encompasses a blend of multiple loss components. I recommend that the authors carry out an ablation study, varying hyperparameters, to assess the impact of each loss on the model's performance.
>
> Equation 10 includes $4$ hyperparameters. Out of that, $2$ are already present in the baseline FATDM. FATDM have optimized their values as $\alpha=0.1$ and $\omega=1$ in their paper. We adopted the same values in our model too. For the other hyperparameters that were introduced by our approach ($\epsilon$ and $\gamma$), we already provide an ablation study in Tables 4 and 5 in the paper.
>
> ***

---

> ### Author Response · Authors · 2023-11-22
> **Rebuttal for Reviewer 3: Part 2**
>
> > 6) The technical novelty of the paper appears somewhat limited. The primary contribution appears to be the utilization of distinct representations for each sensitive attribute.
>
> We introduced the idea of learning two different representations (which are combined at the end) for capturing fairness and generalizability to a target domain. We have introduced a very practical problem of having multiple sensitive attributes across different domains in a domain generalization setting and have proposed a simple and clear solution for the same.  We believe that the simplicity of our method is one of the positive aspects of our method as it is very easy for a researcher/engineer to understand and implement the model in practice.
>
> ***
>
> > 7) In the introduction section (Fig. 1), the authors assert that the proposed method can accommodate the heterogeneity of sensitive attributes across domains. However, in the experimental section, the models seem to have access to all sensitive attributes in all domains, which may contradict the initial claim.
>
> We assume that we have access to a set of possible sensitive attributesinformation on whether an attribute is sensitive is not part of the observations and is specified by the user. Hence we identify a possible set of sensitive attributes and train the model to face any subset of sensitive attributes from the possible set. As the sensitivity of an attribute is user-specified and not present in the observations, at the time of training we combine any set of sensitive attributes with any domain randomly and train the model so that the model is prepared for any new combination of the sensitive attribute. We have emphasized this in the Introduction section of our paper.
>
> ***
> > 8) The paper lacks the provision of code and supplementary documentation, which could significantly enhance clarity and reproducibility. Providing these resources would be beneficial for the reader to understand and replicate the methodology.
>
> We have provided a link to the code along with the rebuttal

---

### Official Review · Reviewer_FctA · 2023-10-30

**Soundness:** 3 good
**Presentation:** 2 fair
**Contribution:** 3 good
**Rating:** 6
**Confidence:** 4

**Summary:**

The paper addresses the challenge of fairness transfer in domain generalization, particularly in contexts where multiple sensitive attributes are present and may vary across domains. Traditional domain generalization methods aim to generalize a model's performance to unseen domains but often ignore the aspect of fairness, especially when multiple sensitive attributes are involved.

The authors propose a novel framework capable of handling fairness with respect to multiple sensitive attributes across different domains, including unseen ones. This is achieved through the development of two types of representations: a domain-invariant representation for generalizing model performance and a selective domain-invariant representation for transferring fairness to domains with similar sensitive attributes. A key innovation of the proposed method is its ability to reduce computational complexity significantly.

**Strengths:**

+ The approach to handle multiple sensitive attributes in domain generalization is innovative and addresses a clear gap in existing literature.
+ The use of real-world datasets for experimentation enhances the practical relevance of the research.
+ Learning two types of representations for generalization and fairness is a thoughtful approach that could have broader applications.
+ Reducing the number of required models from $2^n$ to just one is a significant improvement, making the solution more feasible in practical scenarios.

**Weaknesses:**

- There is a lack of detail on the specific fairness metrics employed and how the trade-off between fairness and accuracy is quantitatively managed.
- While the reduction in model count is impressive, there are no details on the scalability of the approach with respect to the size of the data or the complexity of domain environments.
- The definition of "sensitive attributes" is rather arbitrary and vague - is there any specific reason certain attributes (i.e. smiling) count as sensitive?

**Questions:**

- Could you elaborate on the robustness of your method against different types of distribution shifts compared to existing methods by providing ablation studies?

---

> ### Author Response · Authors · 2023-11-22
> **Rebuttal for Reviewer 2:**
>
> **We thank Reviewer $2$ for their appreciation of our work and their constructive comments. We have posted responses to their queries below.**
>
> > 1) There is a lack of detail on the specific fairness metrics employed and how the trade-off between fairness and accuracy is quantitatively managed.
>
> Our ablation studies in Tables 4, 5, and 6 of the paper show that our model's hyperparameters $\epsilon$ and $\gamma$ can be varied to manage the trade-off between fairness and accuracy quantitatively (on CelebA dataset). In general, we noticed slightly better results for performance when $\gamma$ and $\epsilon$ are high and fairness when $\gamma$ and $\epsilon$ are low.
>
> ***
>
> > 2) While the reduction in model count is impressive, there are no details on the scalability of the approach with respect to the size of the data or the complexity of domain environments.
>
> > a. Scalability of the approach with respect to the size of the data.
> We have performed our experiments on data sizes ranging from $58356$ to $255600$ and show that our model performs well across these ranges.
> Below are more details about the data ranges:
> + *CelebA*. Domain: hair color. No of training samples (domain): $43624$ (black hair), $26984$ (blonde), $37414$ (brown), total: $108022$
> + *MIMIC-CXR-Cardiomegaly disease*. Domain: age. No of training samples (domain): $4312$ (age: Under 40), $15390$ (40-60), $26729$ (60-80), $11925$ (80-100), total: $58356$
> + *MIMIC-CXR-Cardiomegaly disease*. Domain: rotation. No of training samples for each domain: $51120$ ($0^\circ$), $51120$ ($15^\circ$) $\ldots$ $51120$ ($60^\circ$), total: $255600$
>
> > b. Complexity of domain environments.
>  We have shown that our method performs well across different types of domain shifts with different no of sensitive attributes.
> +  *CelebA*. Domain shift: hair color. No. of sensitive attributes: 4 (male, big nose, smiling and young).
> +  *MIMIC-CXR-Cardiomegaly disease*. Domain shift: age. No. of sensitive attributes: 2 (gender and race).
> +  *MIMIC-CXR-Cardiomegaly disease*. Domain shift: image rotations. No. of sensitive attributes: 3 (age, gender and race).
>
> ***
>
> > 3) The definition of "sensitive attributes" is rather arbitrary and vague - is there any specific reason certain attributes (i.e. smiling) count as sensitive?
>
> Technically sensitive attributes are those attributes that are subject to societal bias like age, gender, race, etc. We use experiments on the MIMIC CXR dataset where such attributes are present. However, to have a comprehensive set of experiments, we also decided to run experiments on CelebA which is a popular dataset in the field of fairness [4],[10],[11]. The target attribute for the dataset CelebA is Attractiveness. Past papers [4],[10],[11] have chosen Male as the sensitive attribute due to less no of images corresponding to males versus females in the data. Since we are considering multiple sensitive attributes, we randomly chose 3 other attributes: Big Nose, Smiling, and Young. Also, our intuition was that whether a person is Smiling should not contribute to whether they are Attractive.
>
> [4] Fair Mixup: Fairness Via Interpolation, ICLR 2021 \
> [10] Inclusivefacenet: Improving face attribute detection with race and gender diversity, FAT/ML 2018 \
> [11] Leveling down in computer vision: Pareto inefficiencies in fair deep classifiers, CVPR 2022
>
> ***
>
> > 4) Could you elaborate on the robustness of your method against different types of distribution shifts compared to existing methods by providing ablation studies?
>
> Currently, we only consider covariate shift as the distribution shift in the scope of this paper. We will consider other shifts in a future version of the work.

---

### Official Review · Reviewer_kdx1 · 2023-11-01

**Soundness:** 1 poor
**Presentation:** 2 fair
**Contribution:** 2 fair
**Rating:** 1
**Confidence:** 5

**Summary:**

According to the authors, this work proposes a novel approach to handle multiple sensitive attributes, allowing any combination in the target domain. This approach involves learning two representations: one for general model performance and another for transferring fairness to unseen domains with similar sensitive attributes. The proposed method significantly reduces the model requirement from 2^n to just 1 for handling multiple attributes and outperforms existing methods in experiments with unseen target domains.

**Strengths:**

1. According to the authors, this paper introduces a new setting of fair domain generalization with multiple sensitive attributes.
2. Based on the proposed setting, a comprehensive training approach is given.
3. The paper is easy to follow.

**Weaknesses:**

1. Some statements are over-claimed. In the introduction, "FATDM is the only work that addresses..." this is not true. Several works, other than FATDM, address fairness-aware domain generalization but in various paradigms, such as [1], [2], and [3].
2. Figure 2 is unclear to me. What is the unfairness metric "Mean"? Do you mean "mean difference" or others? How do you define "different level of fairness"? Also, the word "level" should be plural "levels". What is the take-home message when observing the drop in performance, and what is the connection between this drop and multiple attributes? How does this observation relate to various domains?
3. In the second item of contributions in the Introduction, except the problem mentioned in the first contribution, what is the other problem when you say "both problems"?
4. What is the relationship between the target domain \Tilde{d} with source domains? Is the target domain shifted from sources due to covariate shift, too? If not, what assumption do you make on target domains? This lack of clarification and, hence, unclear to me. Besides, giving a brief introduction to covariate shifts is necessary.
5. I doubt the novelty of proposing the setting in multiple sensitive attributes. To me, a dataset with multiple sensitive attributes can be easily converted to one with a single sensitive attribute with multiple categories. For example, as stated in the paper, a sensitivity configuration set \mathcal{C}={[0,0], [0,1], [1,0], [1,1]} can be viewed as a set {1,2,3,4} where a single sensitive attribute with four distinct categorical values.
6. Does data sample x include sensitive attribute c?
7. In Eq.(1), "d'" should be replaced by "d''".
8. How to ensure g_\theta encodes an invariant representation across domains? According to Eq.(5), the loss L_{DG} is defined as the expectation across all source domains. Therefore, it is not convincing to me that the generalization encoder can be generalized to an unseen target domain when a covariate shift occurs.
9. In the fairness encoder, x is concatenated with c. I am wondering how to do it empirically when x is an image while c is one of the annotations of the image. Please explain your experiments for implementation using the CelebA as an example.
10. Speaking of fair machine learning in general, it aims to mitigate spurious correlations between sensitive attributes and model outcomes. Although this work mentions fairness multiple times, it is unclear to me how to mitigate the spurious correlations during training. This work proposes that it "minimize the gap between the domain representations that have the same sensitive attribute configurations and maximize the gap for representations with different sensitive attributes". But this does not ensure unfairness is controllable.

[1] Elliot Creager, Jörn-Henrik Jacobsen, Richard Zemel. Environment Inference for Invariant Learning. ICML 2021.

[2] Changdae Oh, Heeji Won, Junhyuk So, Taero Kim, Yewon Kim, Hosik Choi, Kyungwoo Song. Learning Fair Representation via Distributional Contrastive Disentanglement. ACM SIGKDD 2022.

[3] Chen Zhao, Feng Mi, Xintao Wu, Kai Jiang, Latifur Khan, Christan Grant, Feng Chen. Towards Fair Disentangled Online Learning for Changing Environments. ACM SIGKDD 2023.

**Questions:**

See weaknesses.

---

> ### Author Response · Authors · 2023-11-22
> **Rebuttal for Reviewer 1: Part 1**
>
> **We thank Reviewer 1 for their constructive comments to improve our paper. We provide our responses to their queries below.**
>
> > 1) Some statements are over-claimed. In the introduction, ”FATDM is the only work
> that addresses...” this is not true. Several works, other than FATDM, address fairness-aware domain
> generalization but in various paradigms, such as [1], [2], and [3]
>
> We thank Reviewer 1 for pointing us to papers [1], [2], and [3]. We agree that they are all broadly under fairness-aware domain generalization and we have added these papers to our related works section in the revised version of the paper.
> However, we would like to emphasize the difference between these papers and FATDM. Papers [1] and [2] reduce the correlations between target attributes and sensitive groups and their goal is to improve the test accuracy of the predictions of the worst-case represented sensitive group. (E.g. removing correlations of land birds with land).
> FATDM lies in the line of works like [4], [5], [6] which achieve fairness by reducing bias against any instance of a sensitive group by equalizing the predictions for different instances of the sensitive groups via optimizing metrics like equal opportunity, equalized odds or demographic parity (E.g., predicting with similar true positive and false positives whether gender=Male or Female). [3] optimizes equalized odds and is similar to FATDM, however, their focus is to address the online learning problem.
>
> [1] Environment Inference for Invariant Learning, ICML 2021 \
> [2] Learning Fair Representation via Distributional Contrastive Disentanglement, KDD 2022 \
> [3] Towards Fair Disentangled Online Learning for Changing Environments, KDD 2023 \
> [4] Fair Mixup: Fairness Via Interpolation, ICLR 2021 \
> [5] Equality of opportunity in supervised learning, NeurIPS 2016 \
> [6] Empirical Risk Minimization Under Fairness Constraints, NeurIPS 2018
>
> ***
>
> > 2) Figure 2 is unclear to me. What is the unfairness metric "Mean"? Do you mean "mean difference" or others? How do you define "different level of fairness"? Also, the word "level" should be plural "levels". What is the take-home message when observing the drop in performance, and what is the connection between this drop and multiple attributes? How does this observation relate to various domains?
>
> The unfairness metric Mean is the Mean difference metric (Equation below). We have revised Fig 2 in the paper based on this comment.
>
> By different levels of fairness, we mean that the inclusion of additional sensitive attributes can have additional terms to optimize fairness such that performance may be compromised. We have an example below with gender followed by gender and race.
>
> \begin{equation*}
> \mathrm{Mean Difference(g)} = \frac{1}{N}\sum_{i=1}^{N}\sum_{y \in \mathcal{Y}}((h(\mathbf{z}_i)\mid y, g=\mathrm{F}) - (h({\mathbf{z}_i})\mid y, g=\mathrm{M}))^2
> \end{equation*}
>
> \begin{equation*}
> \begin{split}
> \mathrm{Mean Difference(g,r)} = \frac{1}{N}\sum_{i=1}^{N}\sum_{y \in \mathcal{Y}}((h(\mathbf{z}_i)\mid y, g=\mathrm{F}), r=\mathrm{B})  - (h({\mathbf{z}_i})\mid y, g=\mathrm{M},r=\mathrm{W}))^2 \\ +((h(\mathbf{z}_i)\mid y, g=\mathrm{M}), r=\mathrm{B}) - (h({\mathbf{z}_i})\mid y, g=\mathrm{M},r=\mathrm{W}))^2  \\ +
> ((h(\mathbf{z}_i)\mid y, g=\mathrm{F}), r=\mathrm{B}) - (h({\mathbf{z}_i})\mid y, g=\mathrm{F},r=\mathrm{W}))^2  \\ +
> ((h(\mathbf{z}_i)\mid y, g=\mathrm{M}), r=\mathrm{B}) - (h({\mathbf{z}_i})\mid y, g=\mathrm{F},r=\mathrm{W}))^2  \\ +
> ((h(\mathbf{z}_i)\mid y, g=\mathrm{M}), r=\mathrm{B}) - (h({\mathbf{z}_i})\mid y, g=\mathrm{M},r=\mathrm{B}))^2 \\ +
> ((h(\mathbf{z}_i)\mid y, g=\mathrm{M}), r=\mathrm{W}) - (h({\mathbf{z}_i})\mid y, g=\mathrm{F},r=\mathrm{W}))^2
> \end{split}
> \end{equation*}
> where $\mathbf{z}$ is the latent representation of the input $\mathbf{x}$. \
> **Take home message when observing the drop in performance:**
> Increasing the number of sensitive attributes can compromise the accuracy as shown in Figure 2 in the paper. We assume that an attribute's sensitivity is a user preference and do not expect each domain (of domain generalization) in the data to have the same set of sensitive attributes. When only a subset of attributes (gender or race or none) is sensitive in a domain, enforcing fairness on all possible sets of sensitive attributes (gender and race) can lead to an unnecessary dip in performance.
>
> ***
>
> >3) What is the other problem when you say both problems
>
> The first problem is getting a good generalization performance on a target domain. The second problem is getting a good fairness measure with respect to all the sensitive attributes in the target domain.
>
> ***
>
> > 4) Is the target domain shifted from sources due to covariate shift, too?
>
> yes. However, the sensitive attributes of the target domain are decided by the user. We do not assume it to be due to the covariate shift.
>
> ***

---

> ### Author Response · Authors · 2023-11-22
> **Rebuttal for Reviewer 1: Part 2**
>
> > 5) I doubt the novelty of proposing the setting in multiple sensitive attributes. To me, a dataset with multiple sensitive attributes can be easily converted to one with a single sensitive attribute with multiple categories.
>
> It may be possible to view multiple sensitive attributes as a single sensitive attribute with multiple categories. **However, our contribution is not to just extend domain generalization and fairness to a multi-attribute setting. In a multi-attribute setting, each domain can have a different set of sensitive attributes (which we assume are decided by a user based on the application at hand). Hence, we also address the problem of the target domain having a different set of sensitive attributes from that of the source.** We discuss these contributions under paragraphs $2$ and $3$ of the introduction section of the paper. \
>
> Our baseline is a naive modification of the existing approach FATDM (designed for a single sensitive attribute) to cater to a multi-attribute setting. We compare it against our method SISA which considers the differences in the sensitivity of the attributes across different domains. From the experimental results, we show that our method is better at both performance on the target domain and fairness measures for all subsets from a possible set of sensitive attributes in the target domain.
>
> ***
>
> > 6) Does data sample $\mathbf{x}$ include sensitive attribute $\mathbf{c}$?
>
> No, $\mathbf{x}$ is an image of size $256$x$256$x$1$ for MIMIC CXR and $256$x$256$x$3$ for CelebA. $\mathbf{c}$ is a binary vector of size $n$x$1$ where $n$ is the total number of possible sensitive attributes. We provide it to the model as additional meta information.
>
> ***
>
> > 7) In Eq.(1), "d'" should be replaced by "d''"
>
> No, we use $\mathbf{x}''$ and $\mathbf{x}'$ and $G''$ and $G'$ to denote the differences in the density transformation model and its output. However, both  $\mathbf{x}''$ and $\mathbf{x}'$ are from $d' \in D$.
>
> ***
>
> > 8) How to ensure $g_\theta$ encodes an invariant representation across domains? According to Eq.(5), the loss $L_{DG}$ is defined as the expectation across all source domains. Therefore, it is not convincing to me that the generalization encoder can be generalized to an unseen target domain when a covariate shift occurs.
>
> Prior works [7],[8],[9] have theoretically proved that error on the target domain is upper bounded by the training error on the source domains, pairwise divergences among the source domains, and divergence between the source and the target domain. As we do not have access to target domains, we follow previous works and minimize the pairwise divergence among the source domains through domain invariant representation learning ($L_{DG}$) and the training error on the source domains ($L_{ERR}$) for a lower error on the predictions of the target domain.
>
> [7] Generalizing to unseen domains via distribution matching, 2019 \
> [8]  Fairness and accuracy under domain generalization, ICLR 2023 \
> [9] On Learning Domain-Invariant Representations for Transfer Learning with Multiple Sources, NeurIPS 2021
>
> ***
>
> > 9) In the fairness encoder,  $\mathbf{x}$ is concatenated with $\mathbf{c}$. I am wondering how to do it empirically when $\mathbf{x}$ is an image while $\mathbf{c}$ is one of the annotations of the image. Please explain your experiments for implementation using the CelebA as an example.
>
> $\mathbf{c}$ is a binary vector of size $n$x$1$. We reshape it to $256$x$256$x$1$ by repeating its values over dimensions $1$ and $2$ and adding a $3$rd dimension. Then it is concatenated to input $\mathbf{x}$ as an additional channel. So the total dimension of the input ($\mathbf{x}+\mathbf{c}$ = $256$x$256$x$2$ for MIMIC CXR and $256$x$256$x$4$ for CelebA). We have provided the code to get more insight into this.
>
> ***
>
> > 10) Speaking of fair machine learning in general, it aims to mitigate spurious correlations between sensitive attributes and model outcomes. Although this work mentions fairness multiple times, it is unclear to me how to mitigate the spurious correlations during training. This work proposes that it "minimize the gap between the domain representations that have the same sensitive attribute configurations and maximize the gap for representations with different sensitive attributes". But this does not ensure unfairness is controllable.
>
> Our focus is on the notion of fairness that aims to balance classifier errors across population subgroups, for example, by matching error rates across genders or different racial groups. \
> However, based on the review, we have added Table 13 to the supplementary section of our paper showing the Pearson correlation measure between the sensitive attributes and the target attribute predictions of our model and ERM. Our model reduces the correlation between the sensitive attributes and the model prediction.

---

### Author Response · Authors · 2023-11-22
**Code and trained models**

**We provide an anonymous Google Drive [link](https://drive.google.com/drive/folders/12YnjoixwF-2wCHaonJexbwiDNLQP4_K8?usp=sharing) for sharing the source code and a few trained models for reproducibility.**

---

### Meta-Review · Area_Chair_jUk8 · 2023-12-13

**Metareview:**

This paper received the following ratings: 1, 6, 3, 3.
The main issue, evidenced by all reviewers, regards the poor clarity of the presentation/organization of the paper, even lacking of details, which prevent the full understanding of the work.
Further, the motivations are not well addressed, the novelty aspects are unclear, and experimental validation is insufficient, including poor ablations. Overall, the number of flaws raised by the reviewers are indeed large. Authors provided answers to these issues, but they have not succeeded to convince the reviewers to raise their scores.

In the end, given all these problems, this paper cannot be considered acceptable for publication at ICLR 2024.

**Justification For Why Not Higher Score:**

Too large number of issues and rebuttal was not considered adequate, even if only one reviewer seemed to acknowledge its reading.
Overall, I think this work cannot be accepted in these conditions.

**Justification For Why Not Lower Score:**

N/A

---

### Decision · Program_Chairs · 2024-01-16

Reject